# Peroxisomes form intralumenal vesicles with roles in fatty acid catabolism and protein compartmentalization in Arabidopsis

Zachary J. Wright [1] & Bonnie Bartel [1✉]

Peroxisomes are vital organelles that compartmentalize critical metabolic reactions, such as the breakdown of fats, in eukaryotic cells. Although peroxisomes typically are considered to consist of a single membrane enclosing a protein lumen, more complex peroxisomal membrane structure has occasionally been observed in yeast, mammals, and plants. However, technical challenges have limited the recognition and understanding of this complexity. Here we exploit the unusually large size of Arabidopsis peroxisomes to demonstrate that peroxisomes have extensive internal membranes. These internal vesicles accumulate over time, use ESCRT (endosomal sorting complexes required for transport) machinery for formation, and appear to derive from the outer peroxisomal membrane. Moreover, these vesicles can harbor distinct proteins and do not form normally when fatty acid β-oxidation, a core function of peroxisomes, is impaired. Our findings suggest a mechanism for lipid mobilization that circumvents challenges in processing insoluble metabolites. This revision of the classical view of peroxisomes as single-membrane organelles has implications for all aspects of peroxisome biogenesis and function and may help address fundamental questions in peroxisome evolution.

[1] Biosciences Department, Rice University, Houston, TX, USA. ✉email: bartel@rice.edu

Eukaryotic life is distinguished by compartmentalizing biochemical reactions in membrane-delimited organelles, and how organelles accomplish this sequestration is a fundamental question of cell biology. Peroxisomes are organelles that house vital oxidative reactions, such as fatty acid β-oxidation, and are necessary for life in most multicellular eukaryotes, including plants[1]. Despite their discovery in the mid-twentieth century[2] and their importance in cancer[3], aging[4], and fatal human peroxisome biogenesis disorders[5], many basic processes governing peroxisome structure and dynamics remain incompletely characterized.

Peroxisomes can form by fission of preexisting organelles or through de novo formation[6], in which pre-peroxisomes bud from the endoplasmic reticulum (ER) and become functional after protein acquisition. Lumenal (matrix) proteins are encoded in the nucleus and post-translationally imported into peroxisomes following recognition by cytosolic receptors of a C-terminal peroxisomal targeting signal 1 (PTS1) or an N-terminal PTS2. Peroxisomal membrane proteins (PMPs) are inserted via a membrane PTS (mPTS), either directly (PMPs synthesized in the cytosol and accompanied by a chaperone to the peroxisome membrane for insertion[7]) or indirectly (PMPs transit through the ER to peroxisome membranes[6,8]).

*Arabidopsis thaliana* is a well-established platform for investigating peroxisome biology[1] that is uniquely suited for probing peroxisome structure. Arabidopsis peroxisomes are relatively large (~1–2 μm) and can be even larger in young seedlings[9]. Moreover, Arabidopsis is amenable to live-cell imaging, and individual peroxisomes can be followed for hours with high spatial resolution.

In this work, we exploit the advantages of the Arabidopsis system to investigate peroxisome membrane structure and dynamics. We use fluorescent reporter proteins that simultaneously label peroxisomal membranes and lumen to reveal abundant intralumenal vesicles (ILVs) within peroxisomes. We begin to dissect the molecular requirements for this unexpected complexity and uncover roles for these structures in protein compartmentation and lipid mobilization.

## Results

**Reporters reveal inner peroxisomal membranes**. To visualize peroxisomes, we fused bright, monomeric fluorescent reporter proteins to the targeting sequences from two peroxins (PEX proteins, the proteins necessary for peroxisome biogenesis) that function at the peroxisomal membrane. We generated a tail-anchored reporter with mNeonGreen fused to the C-terminal PEX26 mPTS[7] and an N-terminally anchored reporter with the N-terminal PEX22 mPTS[10] fused to mNeonGreen (Supplementary Fig. 1a). We stably transformed Arabidopsis with constructs constitutively expressing these reporters along with mRuby3-PTS1 to label the peroxisome lumen (Supplementary Fig. 1a) and visualized peroxisomes in living plants using confocal microscopy.

The mRuby3-PTS1 reporter labels the peroxisomal lumen, and we detected abundant mRuby3-PTS1 puncta in seedlings (Fig. 1a, b). mNeonGreen-mPTS$^{PEX26}$ localized with mRuby3-PTS1 fluorescence (Fig. 1c), consistent with peroxisome labeling, whereas mPTS$^{PEX22}$-mNeonGreen labeled both peroxisomes and reticular membranes resembling the ER (Fig. 1e). These distinct localizations suggest that tail-anchored mNeonGreen-mPTS$^{PEX26}$ inserts directly into peroxisome membranes whereas N-terminally anchored mPTS$^{PEX22}$-mNeonGreen inserts into the ER before or in addition to sorting to peroxisomes.

Imaging 8-day-old seedling peroxisomes with our dual reporters did not reveal the expected well-defined membrane enclosing a clear lumen (Fig. 1c–f). Instead, the membrane reporters appeared throughout the organelle, often completely (Fig. 1d) or nearly (Fig. 1f) filling the peroxisomal lumen. We occasionally observed larger peroxisomes that seemed to have more complex morphology (Fig. 1c). Because this distribution was unexpected, we examined younger seedling peroxisomes, which can be much larger while seed lipid stores are being catabolized by β-oxidation[9]. We found that some younger seedling peroxisomes also had the membrane reporter localized uniformly throughout the lumen (Fig. 1g). In contrast, larger peroxisomes revealed both membrane reporters labeling not only the delimiting organellar membrane, but also numerous internal vesicles (Fig. 1g–j and Supplementary Movies 1, 2). These ILVs were present in all peroxisomes large enough to have a resolved lumen and varied in number, size, and morphology. Some peroxisomes had a few large vesicles, some had numerous smaller vesicles, and some had both large and small vesicles (Fig. 1g–j). Additionally, some internal peroxisome membranes resembled tubules (Fig. 1k), reminiscent of the ILV concatenation sometimes detected in plant endosomes[11].

The unexpected ubiquity of peroxisomal ILVs prompted us to confirm that our reporters were labeling peroxisomes. To determine whether our peroxisome membrane reporter was mislocalizing to multivesicular endosomes, organelles known to harbor internal vesicles, we used the fluorescent dye FM4-64, which binds to the plasma membrane and marks endocytosed membranes over time. Our peroxisome membrane reporter labeled organelles distinct from FM4-64-labeled endosomes (Fig. 1l and Supplementary Movie 3). We also examined whether we were observing autophagy-related structures by examining mNeonGreen-mPTS$^{PEX26}$ in *atg7-4*, which lacks an enzyme required for autophagosome formation[12]. The presence and pervasiveness of peroxisomal ILVs appeared unchanged when autophagy was prevented (Fig. 1m and Supplementary Movie 4), confirming that our reporter was not labeling autophagosomes and that peroxisomal ILVs do not result from an autophagic process.

To determine whether our membrane-targeted fusion proteins aberrantly triggered ILV formation, we examined the mRuby-PTS1 reporter in lines without a membrane reporter. We detected spherical nonfluorescent regions in the peroxisomal lumen (Fig. 1n and Supplementary Movie 5), indicating that peroxisomal ILVs were not an artifact of our membrane reporters.

**Peroxisomes densely pack ILVs over time**. The striking range of peroxisome sizes and inner membrane densities observed (Fig. 1a–j) prompted us to examine ILVs over time. Seed oil reserves allow young Arabidopsis seedlings to survive for days without exogenous nutrients or sunlight, facilitating long-term live-cell imaging[13]. We observed large (9–12 μm diameter) peroxisomes in 4-day-old seedlings shrinking over several hours while acquiring additional ILVs (Fig. 2 and Supplementary Movies 6, 7). In contrast, the size of smaller peroxisomes (<5 μm) did not appreciably change during the imaging period (Fig. 2 and Supplementary Movies 6, 7). Over time, some peroxisomes became so densely packed that the membrane reporter appeared uniform throughout the lumen, and we could no longer resolve individual ILVs (Fig. 2a, b). The decrease in peroxisome size as internal membranes formed (Fig. 2) implies that ILVs are derived from the outer peroxisomal membrane and that smaller peroxisomes in older seedlings (Fig. 1a–f) may arise from larger peroxisomes (Fig. 1g–j) becoming increasingly packed with inner membranes over time.

**ESCRT proteins contribute to peroxisomal ILV formation**. Seeking mechanistic understanding of peroxisomal ILV formation, we examined the role of the endosomal sorting complexes

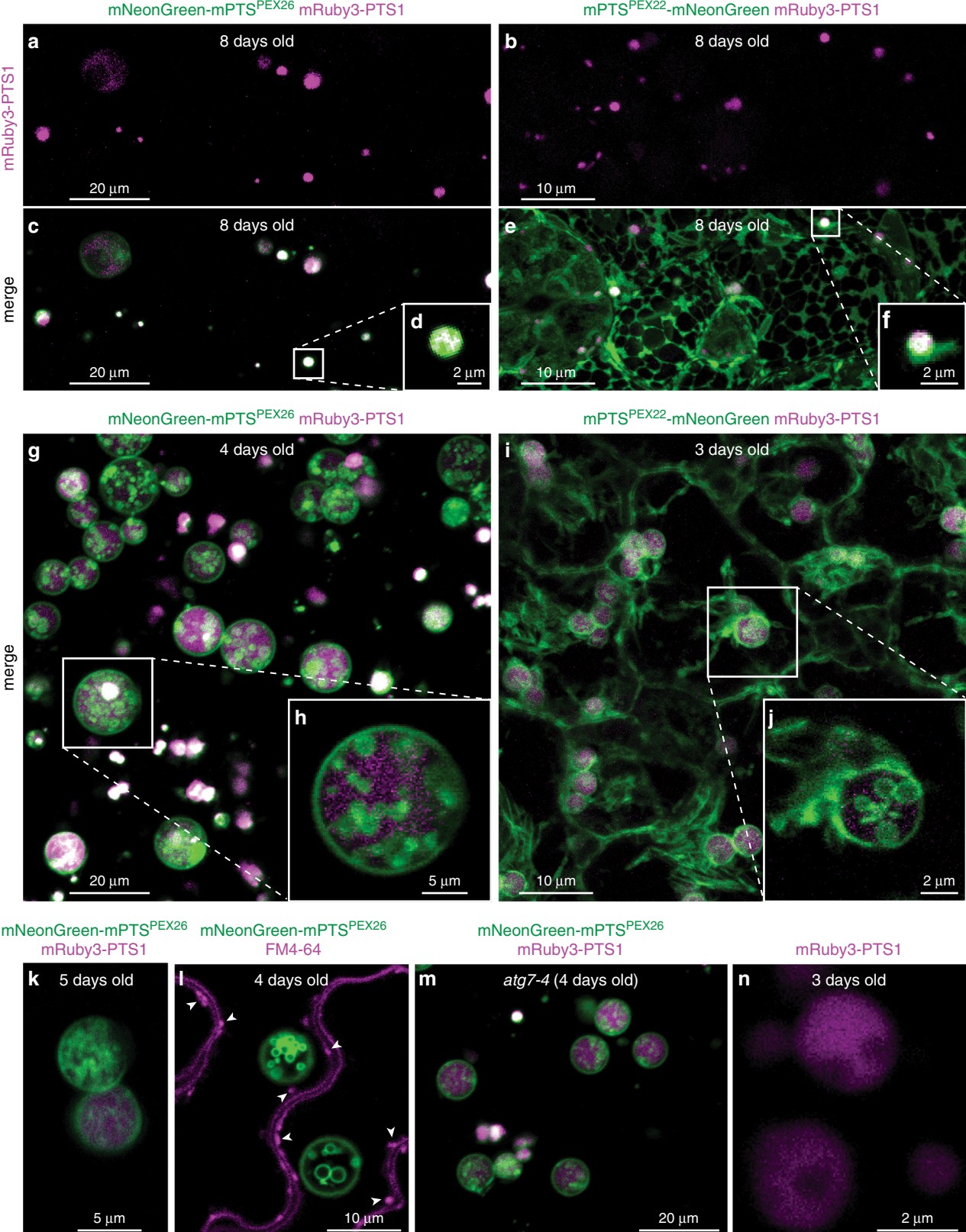

required for transport (ESCRT). During endosomal ILV formation, ESCRT proteins corral selected membrane proteins into membrane invagination sites and physically deform the membrane into an ILV that is pinched off into the endosomal lumen[14,15]. We therefore tested whether ESCRT machinery participated in peroxisomal ILV formation.

Because completely disabling ESCRT functions in Arabidopsis is lethal[16], we expressed inducible dominant-negative ESCRT variants[16,17]. We generated β-estradiol-inducible[18] constructs (Fig. 3a) to express either SNF7[L22W], which interferes with a filament-forming ESCRT-III component, or VPS4[E232Q], an inactivate version of the ATPase that provides mechanical force

**Fig. 1 Membrane reporters reveal inner peroxisomal membranes that are not endosomes or autophagosomes. a, b** Z-projections of peroxisomes imaged using mRuby3 channel (magenta) of 8-day-old seedling cotyledon epidermal cells expressing mNeonGreen-mPTS[PEX26] (**a**) or mPTS[PEX22]-mNeonGreen (**b**) and mRuby3-PTS1. **c** Merged mNeonGreen-mPTS[PEX26] (green) and mRuby3-PTS1 (magenta) channels from (**a**). **d** Cross-section of peroxisome boxed in (**c**). **e** Merged mPTS[PEX22]-mNeonGreen (green) and mRuby3-PTS1 (magenta) channels from (**b**) reveals peroxisomes and fluorescence resembling ER. **f** Cross-section of peroxisome boxed in (**e**). **g** Z-projection of 4-day-old cotyledon epidermal cells expressing mNeonGreen-mPTS[PEX26] (green) and mRuby3-PTS1 (magenta) show ILVs in all peroxisomes with a resolvable lumen. **h** Cross-section of peroxisome boxed in (**g**). **i** Z-projection of 3-day-old cotyledon epidermal cells expressing mPTS[PEX22]-mNeonGreen (green) and mRuby3-PTS1 (magenta) shows ILVs in peroxisomes and extra-peroxisomal fluorescence resembling ER. **j** Cross-section of peroxisome boxed in (**i**). **k** Cross-section of cotyledon cell expressing mPTS[PEX26]-mNeonGreen (green) and mRuby3-PTS1 (magenta) showing inner peroxisomal membranes with tubular morphology. **l** Cross-section of cotyledon epidermal cells expressing mNeonGreen-mPTS[PEX26] (green) and stained with FM4-64 (magenta), which marks plasma membrane and endosomes (e.g., arrowheads). **m** Z-projection of cotyledon epidermal cells of *atg7-4* expressing mNeonGreen-mPTS[PEX26] (green) and mRuby3-PTS1 (magenta) showing abundant peroxisomal ILVs remain when autophagy is prevented. **n** Cross-section of hypocotyl cells expressing mRuby3-PTS1 (magenta) without a membrane reporter showing nonfluorescent "holes" in peroxisomes.

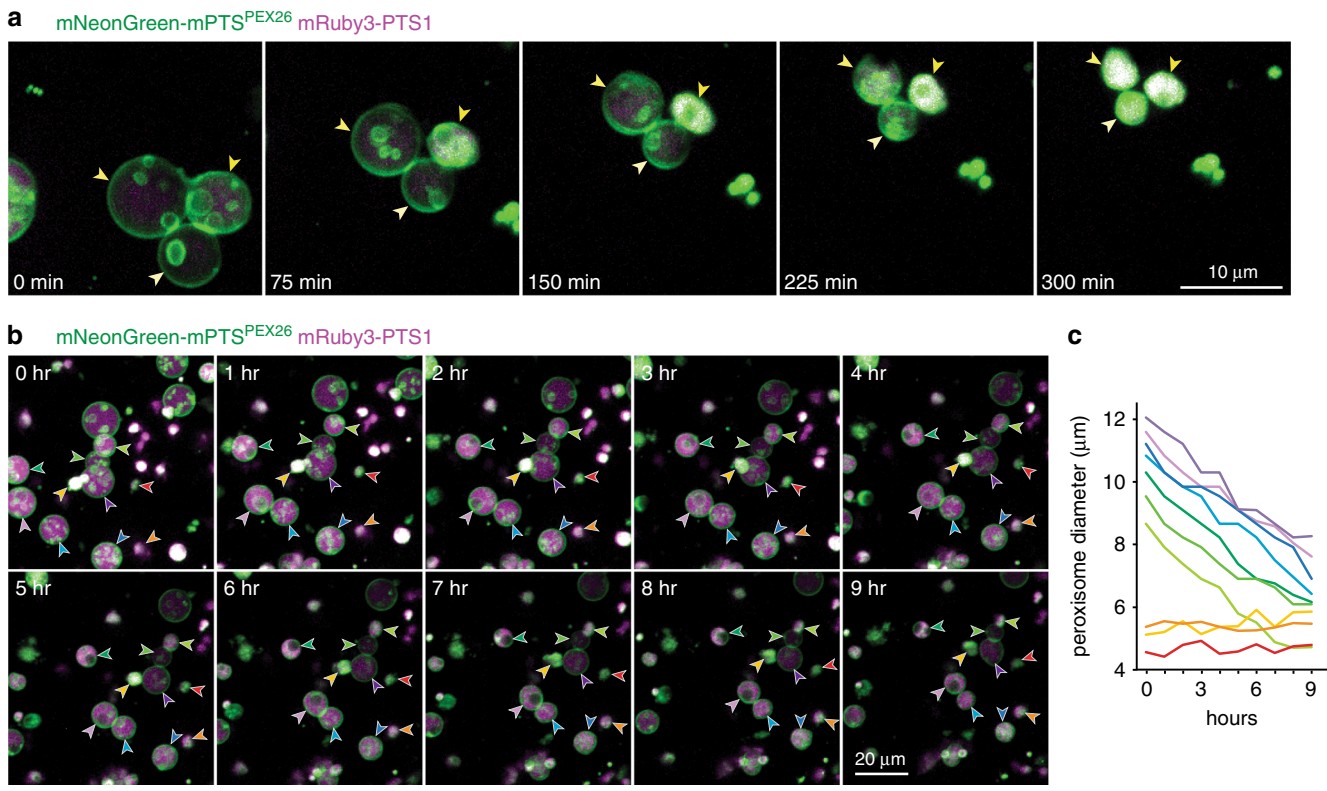

**Fig. 2 Peroxisome size decreases as inner membrane density increases. a, b** Z-projections of cotyledon epidermal cells of a 4-day-old seedling expressing mNeonGreen-mPTS[PEX26] (green) and mRuby3-PTS1 (magenta) imaged with high-magnification spinning disk confocal microscopy over 5 h (**a**) or lower-magnification scanning confocal microscopy over 9 h (**b**). Individual peroxisomes are designated with differently colored arrowheads. **c** Diameters of individual peroxisomes marked with arrowheads of same color in panel (**b**).

for membrane scission[16,19]. We selected transformants that accumulated detectable dominant-negative protein only in the presence of β-estradiol (Fig. 3b), crossed these lines to our mNeonGreen-mPTS[PEX26] mRuby3-PTS1 dual reporter, and imaged peroxisomes in 5-day-old seedlings, when most peroxisomes in wild type have acquired ILVs and decreased in size.

Expressing either dominant-negative ESCRT protein altered peroxisome morphology. In β-estradiol-treated seedlings expressing dominant-negative ESCRT components, many peroxisomes were larger than in control seedlings (Fig. 3c, d), suggesting that ILV formation from the outer peroxisomal membrane was impaired. Despite these morphological defects, ESCRT disruption did not detectably impede lumenal protein import; mRuby3-PTS1 remained peroxisomal (Fig. 3c), and processing of an endogenous PTS2 protein, which occurs after import[1], remained robust (Fig. 3b).

**Fatty acid β-oxidation proteins influence ILV formation.** Having implicated ESCRT proteins in peroxisomal ILV formation, we sought peroxisome-specific components of ILV biogenesis. The peroxisome shrinkage we observed as ILVs formed (Fig. 2) suggested that mutants with enlarged peroxisomes might have ILV formation defects. Several peroxisome-defective mutants display enlarged peroxisomes[9,20,21], including *pxn-4*, a mutant of the peroxisomal ATP and NAD[+] transporter; *acx2-2*, a mutant in the very-long-chain acyl-CoA oxidase; and *mfp2-8*, a mutant in the long-chain isoform of the multifunctional protein[9]. To test whether these mutations altered peroxisomal ILV formation, we examined our peroxisome reporters in these mutants (Fig. 4a). As previously reported[9], peroxisomes in wild-type seedlings started small, enlarged 3−4 days after sowing, and then became mostly small again, while peroxisomes in the mutant seedlings started small but continued to enlarge; in older mutant

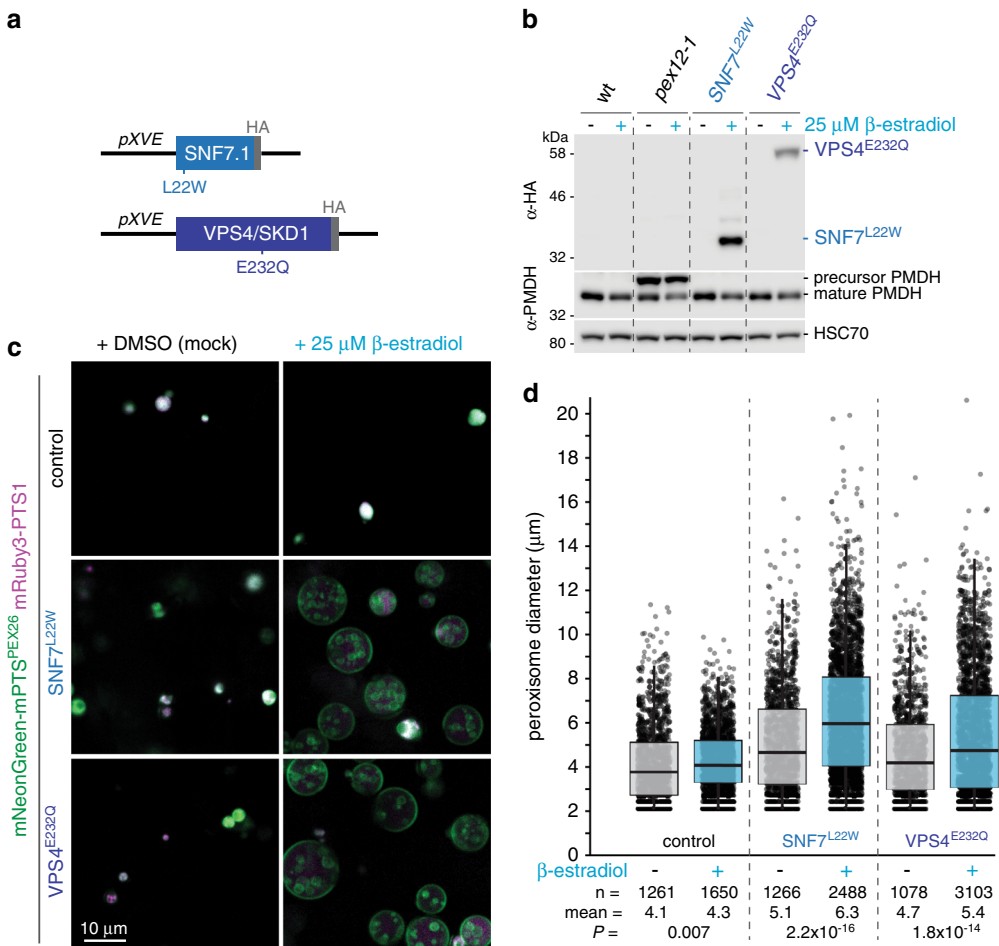

**Fig. 3 ESCRT function is required for efficient peroxisomal inner membrane formation. a** Diagram of dominant-negative ESCRT constructs driven by the β-estradiol-inducible *pXVE* promoter system[18]. **b** Dominant-negative ESCRT derivatives exhibit estradiol-dependent accumulation and do not impair peroxisomal protein processing. Immunoblots of 5-day-old seedlings grown without or with 25 μM β-estradiol are shown. The PTS2 signal region of the PMDH precursor is cleaved from mature PMDH in the peroxisome lumen. Precursor accumulation in the *pex12-1* mutant[79] reflects protein import defects. **c** ESCRT disruption impairs peroxisomal ILV formation. Cotyledon epidermal cells are shown of 5-day-old seedlings expressing mNeonGreen-mPTS$^{PEX26}$ (green) and mRuby3-PTS1 (magenta) and lacking (control) or carrying constructs expressing dominant-negative SNF7$^{L22W}$ or VPS4$^{E232Q}$ grown with DMSO (mock) or β-estradiol. **d** ESCRT disruption causes enlarged peroxisomes. Peroxisome diameter was quantified from images of four seedlings (Supplementary Fig. 2) expressing mNeonGreen-mPTS$^{PEX26}$ and mRuby3-PTS1 and grown without (gray) or with (blue) 25 μM β-estradiol to induce SNF7$^{L22W}$ or VPS4$^{E232Q}$ expression. Boxes show second and third quartiles, horizontal lines are medians, and whiskers encompass data points up to 1.5 times the interquartile range past the box ends. Student's two-tailed unpaired *t* tests were used to compare peroxisome size with and without β-estradiol.

seedlings some peroxisomes were over 30 μm in diameter, approaching the size of the large central vacuole. Additionally, whereas wild-type peroxisomes had abundant inner membranes, such that the membrane reporter often largely overlapped the lumenal reporter, the mutants often had less densely packed peroxisomes, with the largest peroxisomes often having few or no ILVs (Fig. 4a, c). Moreover, when inner membranes were present in mutant peroxisomes, we often observed membrane aggregations or vesicle clumps attached to the lumenal face of the outer membrane, unlike the predominantly free-floating vesicles observed in wild-type peroxisomes (Fig. 4c and Supplementary Movies 8, 9). These apparent sites of disrupted ILV formation suggest that these β-oxidation proteins support inner membrane formation.

Finding ILV-deficient peroxisomes in mutants with impaired fatty acid β-oxidation prompted us to examine ILV formation when fatty acid transport into peroxisomes was disrupted. Fatty acids are imported into peroxisomes by the ABC transporter PXA1[22,23], and the large-peroxisome phenotype of certain β-oxidation mutants can be suppressed by disrupting PXA1[9]. We

examined our reporters in *pxa1-1*, a splice-site mutant missing the last 32 amino acids of PXA1[22], and in the *pxn-4 pxa1-1* double mutant[9]. As previously reported[9], peroxisomes in *pxa1-1* remained small throughout seedling development and the large peroxisomes of *pxn-4* were not present in lines also carrying the *pxa1-1* lesion (Fig. 4a, b). Closer examination, particularly in root tissue, revealed that *pxa1-1* and *pxn-4 pxa1-1* peroxisomes often lacked most inner membranes and instead presented the classical peroxisome morphology of a single membrane enclosing a protein-only lumen (Fig. 4d).

**Peroxisomal ILVs may assist fatty acid import**. The observations that not only peroxisome enlargement[9] but also ILV formation (Fig. 4) during early seedling development depend on fatty acid import suggested that peroxisomal ILVs might function in fatty acid import. The fatty acids that fuel seedling germination are stored as triacylglycerol (TAG) in lipid droplets[1]. To further illuminate the linkage between ILV formation and fatty acid import, we examined whether impeding ESCRT function, which

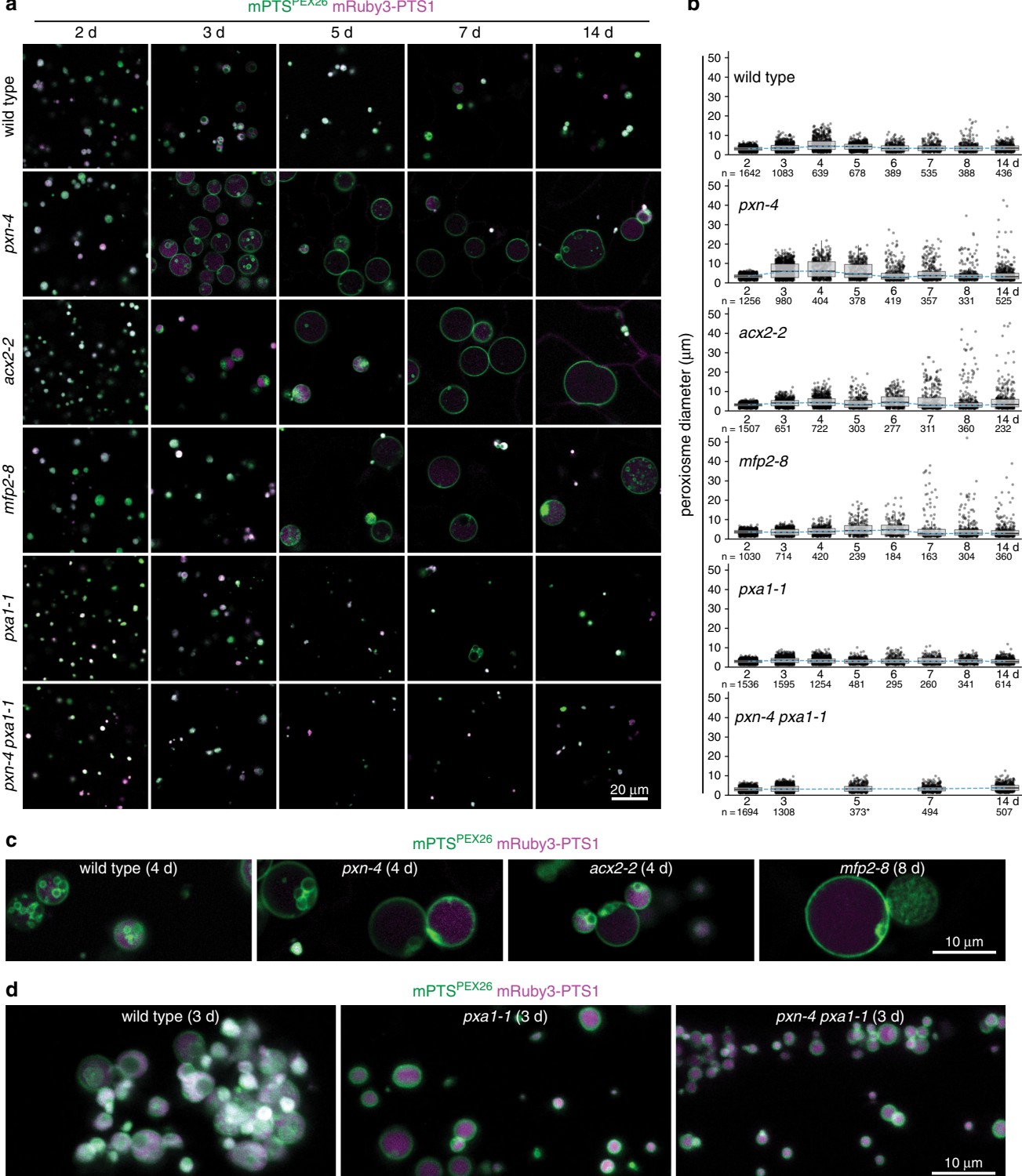

**Fig. 4 Efficient peroxisomal ILV formation requires β-oxidation. a**, **b** Enlarged peroxisomes in mutants with impaired β-oxidation show disrupted ILV formation. Wild-type seedlings and various mutants with impaired β-oxidation expressing mNeonGreen-mPTS$^{PEX26}$ (green) and mRuby3-PTS1 (magenta) were imaged over several days; cross-sections (**a**) and peroxisome diameters (**b**) are shown. Boxes show second and third quartiles, horizontal lines are medians, and whiskers encompass data points up to 1.5 times the interquartile range past the box ends. **c** Magnified peroxisome cross-sections show ILVs in wild-type peroxisomes dispersed in the lumen and residual ILVs in *pxn-4*, *acx2-2*, and *mfp2-8* aggregated near the outer membrane. **d** Z-projections of peroxisomes in root cells show wild-type peroxisomes with densely packed inner membranes and *pxa1-1* and *pxn-4 pxa1-1* peroxisomes lacking most inner membranes.

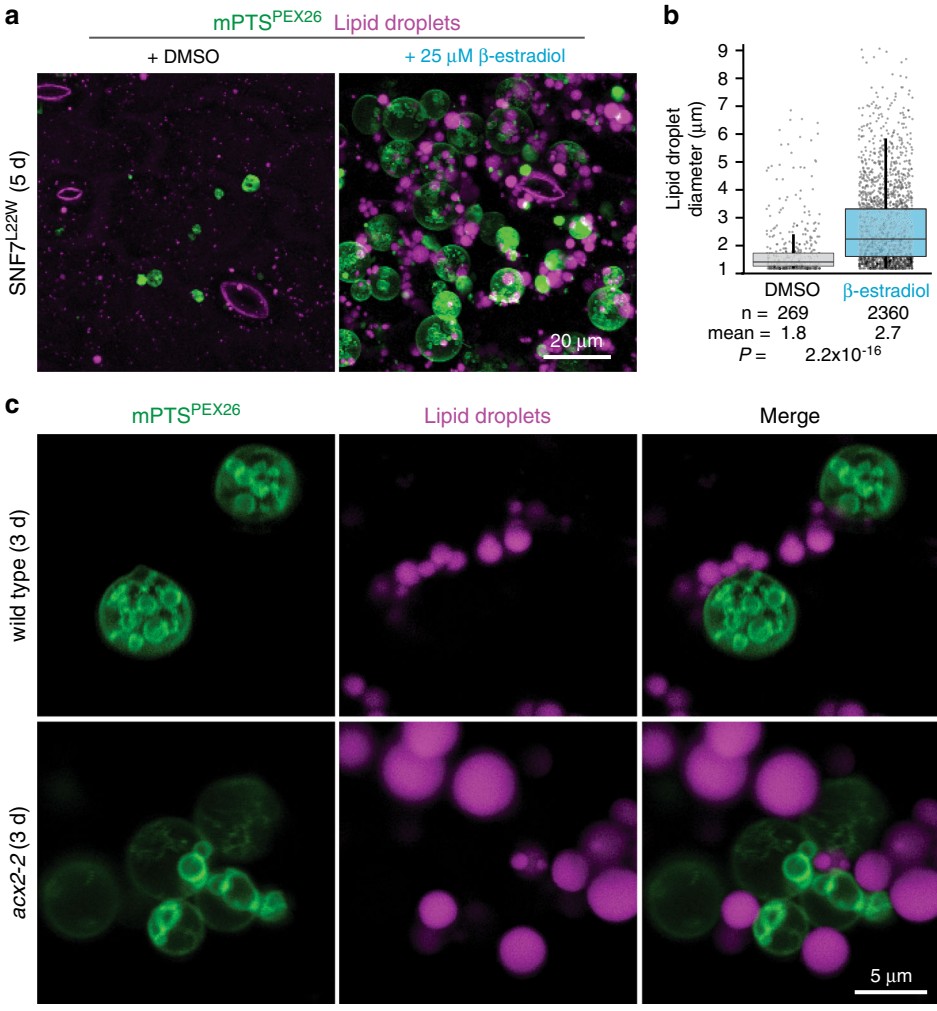

**Fig. 5 Peroxisomal vesicles lack engulfed lipid droplets. a** Dominant-negative ESCRT induction impairs lipid mobilization. *Z*-projections of cotyledon epidermal cells of seedlings expressing mNeonGreen-mPTS$^{PEX26}$ (green) and mRuby3-PTS1 stained with monodansylpentane (MDH) to mark lipid droplets (magenta) and grown with DMSO (mock) or β-estradiol to induce SNF7$^{L22W}$ expression. **b** Lipid droplet diameter in SNF7$^{L22W}$ seedlings from panel (**a**). Boxes show second and third quartiles, horizontal lines are medians, and whiskers encompass data points up to 1.5 times the interquartile range past the box ends. Student's two-tailed unpaired *t* tests were used to compare lipid droplet size with and without β-estradiol. **c** Peroxisomal ILVs do not accumulate neutral lipids. Cross-section cotyledon epidermal cells from wild-type or *acx2-2* expressing mNeonGreen-mPTS$^{PEX26}$ (green) and lipid droplets stained with MDH (magenta).

disrupts ILV formation (Fig. 3c), also impaired seedling fatty acid mobilization. We measured lipid droplets following mono-dansylpentane (MDH) staining of 5-day-old seedlings, after most seed lipid stores have been metabolized in wild type, with and without induction of the SNF7$^{L22W}$ dominant-negative ESCRT variant. We observed substantially larger lipid droplets following ESCRT impairment (Fig. 5a, b), supporting the hypothesis that peroxisomal ILV formation might promote lipid mobilization.

To examine whether peroxisomal ILVs facilitated lipid transport into peroxisomes by engulfing lipid droplet blebs, we stained 4-day-old seedlings expressing the peroxisomal membrane reporter with MDH. We found that peroxisomal ILVs lacked MDH-stained lipids (Fig. 5c and Supplementary Movie 8), indicating that these vesicles are not engulfed lipid droplet blebs and that any free lipids in the ILV lumen are relatively low in abundance. Additionally, we noted that the aggregated membranes in the *acx2-2* mutant, which displays impaired ILV formation (Fig. 4a, c), were often directly apposed to a lipid droplet (Fig. 5c and Supplementary Movie 9), and unlike wild-type peroxisomes, which have free-floating ILVs (Supplementary Movie 8), these aggregates remained associated with the membrane (Supplementary Movie 9). This spatial relationship between lipid droplets and presumptive sites of ILV invagination supports the hypothesis that fatty acids freed from TAG might partition into peroxisomal ILV membranes to await peroxisomal β-oxidation.

**Peroxisomal ILVs can compartmentalize proteins**. Although inner peroxisomal membranes were most easily visualized in young seedlings (where peroxisomes are large and β-oxidation activity is high), we also observed abundant inner membranes in peroxisomes of older seedlings (Fig. 1a–f), suggesting functions beyond fatty acid import. Interestingly, only a subset of ILVs contained mRuby3-PTS1 (Figs. 1n, 6a), suggesting that ILVs might compartmentalize different lumenal proteins.

To test this hypothesis, we examined SNOWY COTYLEDON3 (SCO3) and a protein of unknown function (UP9), which localize in peroxisomal subdomains compared to a generic PTS1 reporter[24,25]. We examined plants expressing mRuby3-PTS1, mPTS-fused mNeonGreen, and mTagBFP2 fused to UP9 or SCO3 (Supplementary Fig. 1c). Although the SCO3 and UP9

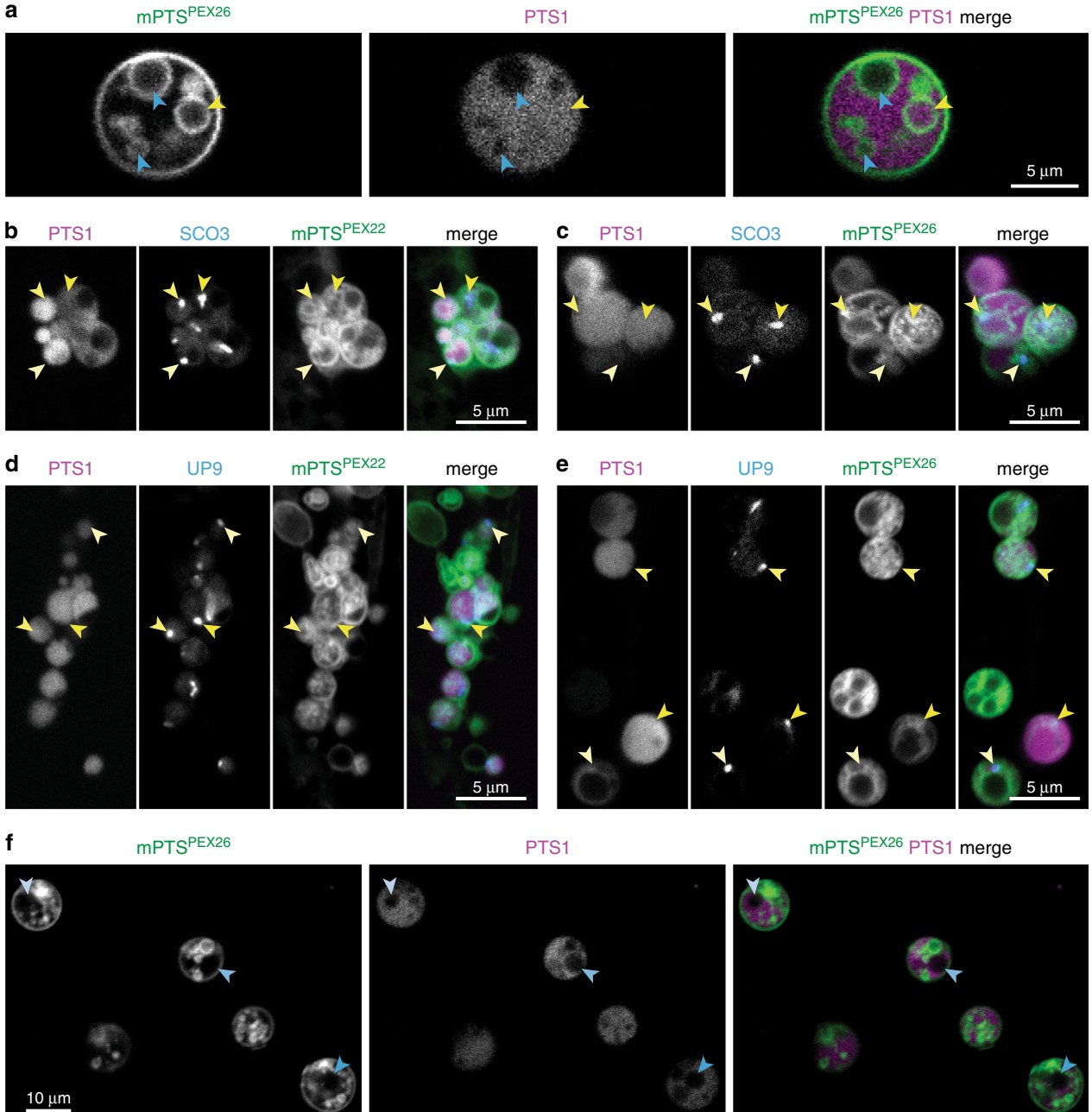

**Fig. 6 Peroxisomal vesicles compartmentalize proteins. a** Cross-section of 4-day-old cotyledon epidermal cell expressing mNeonGreen-mPTS$^{PEX26}$ (green) and mRuby3-PTS1 (magenta) shows peroxisomal ILVs enclosing (yellow arrowhead) or lacking (blue arrowheads) PTS1 reporter. **b, c** Cross-sections of 4-day-old cotyledon epidermal cells expressing mRuby3-PTS1 (magenta), mTagBFP2-SCO3 (blue), and mNeonGreen-mPTS$^{PEX22}$ (green) (**b**) or mPTS$^{PEX26}$-mNeonGreen (green) (**c**) show SCO3 accumulation (arrowheads) in a subset of ILVs. **d, e** Cross-sections of 4-day-old cotyledon epidermal cells expressing mRuby3-PTS1 (magenta), mTagBFP2-UP9 (blue), and mNeonGreen-mPTS$^{PEX22}$ (green) (**d**) or mPTS$^{PEX26}$-mNeonGreen (green) (**e**) show UP9 accumulation (arrowheads) in a subset of ILVs. **f** Cross-section of 3-day-old cotyledon epidermal cells expressing mNeonGreen-mPTS$^{PEX26}$ and mRuby3-PTS1 (magenta) shows ILVs with (green) and without (arrowheads) mNeonGreen-mPTS$^{PEX26}$ membrane fluorescence.

reporters partially colocalized with mRuby3-PTS1, they predominantly concentrated in peroxisomal ILVs where mRuby3-PTS1 was not abundant (Fig. 6b–e). Moreover, some peroxisomal ILVs lacked concentrated mTagBFP2-SCO3 and mTagBFP2-UP9 (Fig. 6b–e), suggesting that lumenal proteins might sort into multiple domains.

Additionally, we detected vesicles (holes in mRuby3-PTS1 fluorescence) that lacked membrane reporter (Fig. 6f and Supplementary Movie 10), suggesting the possibility of distinct PMP localizations. These discrete lumenal and membrane protein localizations (Fig. 6) hint that peroxisomal ILVs may be compartments with distinct protein compositions.

## Discussion

Using bright, monomeric PMP-based reporters, we detected abundant internal membranes in Arabidopsis peroxisomes (Fig. 1). Previous hints of internal peroxisomal membranes have been largely ignored or unexplored. Although peroxisomes in yeast and mammalian cells are usually too small to resolve the

lumen using fluorescence microscopy, internal peroxisomal membranes are occasionally visible in electron micrographs of wild-type yeast[26] and mammalian cells[27] and have been reported in mutants of yeast[28], mammals[29], and plants[30] with large and less densely packed peroxisomes. Notably, some of the earliest electron microscopic visualizations of plant peroxisomes revealed extensive invaginations and internal vesicles within seedling peroxisomes[31–34]. The dramatic morphological changes that we observed as young seedling peroxisomes formed ILVs (Fig. 2 and Supplementary Movie 6) could explain why internal membranes have not been more readily recognized in the past: electron microscopic images of peroxisomes with clearly resolved vesicles might be mistaken for other organelles, and membranes in older peroxisomes might be too densely packed to resolve individual vesicles.

The ESCRT machinery is well known for sculpting ILVs to form multivesicular endosomes[14] and also coordinates other membrane events, including lysosome membrane repair[35], phagophore closure[36], cellular abscission[37], and severing nascent yeast pre-peroxisomal membranes from the ER[38]. Identifying ESCRT machinery partners on the peroxisome membrane would assist in disentangling direct versus indirect roles for ESCRT in peroxisomal ILV formation. Intriguingly, some plant viruses evade host defenses by replicating in ESCRT-derived invaginations of peroxisome membranes[39]. Although viral recruitment of ESCRT proteins to peroxisomes was thought to be aberrant, our discovery of a role for ESCRT in peroxisomal ILV formation in uninfected plants (Fig. 3) suggests that these viruses exploit a normal peroxisomal process. Additionally, our finding that ESCRT is needed for efficient lipid droplet mobilization (Fig. 5a, b) is consistent with a reported role for ESCRT in lipid transfer to peroxisomes in mammals[40].

Peroxisomal ILVs allow a complex distribution of lumenal and membrane proteins (Fig. 6) that may explain reports of sub-peroxisomal protein distribution. For example, although localization heterogeneity can reflect protein crystals or aggregates, some lumenal proteins concentrate in peroxisomal subsections outside of putative aggregates[41]. Additionally, superresolution microscopy of mammalian peroxisomes reveals sub-peroxisomal domains of membrane proteins[42,43]. The sorting pathways directing membrane and lumenal proteins to peroxisomal ILVs remain to be elucidated.

Although extant peroxisomes host diverse metabolic reactions that vary across organisms and tissues, β-oxidation is the earliest traceable metabolic pathway in peroxisomes[44], suggesting that peroxisomes evolved to handle this process. After peroxisomal β-oxidation arose, presumably via retargeting of mitochondrial enzymes[44], mitochondrial β-oxidation was lost in yeast and plants[45]. Interestingly, animals retain β-oxidation in both peroxisomes and mitochondria. Animal peroxisomes preferentially catabolize long (16–20 C)- and very long-chain fatty acids (>20 C), which after shortening are conjugated to carnitine and transferred to mitochondria, where short (<8 C)-, medium (8–14 C)-, and long-chain fatty acids are catabolized[46]. Although both peroxisomes and mitochondria make extensive contacts with lipid droplets, mitochondria appear to move away from lipid droplets in TAG-mobilizing conditions[47]. In contrast, peroxisomes rapidly migrate to lipid droplets during fasting in *C. elegans* and mice[48], and peroxisome-lipid droplet tethers mediate fatty acid channeling for β-oxidation[40]. Thus, it appears that peroxisomes host initial lipid droplet-derived fatty acid catabolism, before shortened fatty acids are passed to mitochondria for completed β-oxidation in animals[49].

The defects in peroxisomal ILV formation that we observe when fatty acid import or β-oxidation is impaired (Fig. 4) suggest that ILVs might help import fatty acids. Perhaps peroxisomes

overcome the challenges of transporting insoluble fatty acids across a lipid bilayer by incorporating free fatty acids into the bilayer and then bringing fatty-acid-enriched vesicles into the organelle. In this model, fatty acids freed from TAG at the lipid droplet−peroxisome interface are delivered to PXA1, which may act as a flippase to move the fatty acid carboxyl group to the lumenal leaflet of the bilayer[50]; the peroxisome would enlarge to accommodate the additional lipids. The ESCRT machinery may invaginate this section of the membrane, and fatty acid β-oxidation enzymes may stabilize the invaginating ILV on the lumenal side. Directly β-oxidizing long-chain fatty acids imbedded in the membrane avoids the free energy cost of pulling the fatty acid out of the lipid bilayer prior to catabolism. Indeed, some long-chain isoforms of fatty acid-metabolizing enzymes preferentially act on fatty acids in a membrane, rather than in solution[51]. When fatty acids become sufficiently soluble to partition into the aqueous lumen, the medium- and short-chain isoforms would complete β-oxidation in the peroxisome in plants and yeast, whereas in animals these now soluble fatty acids would be transported to mitochondria for continued β-oxidation. ILV membranes depleted of free fatty acids might be either recycled to the outer membrane or hydrolyzed to generate additional fatty acids for β-oxidization. Indeed, the yeast peroxisomal phospholipase LPX1 is induced in conditions that promote β-oxidation, and *lpx1* mutants accumulate intraperoxisomal vesicles[52], suggesting that peroxisomes can degrade ILV membranes.

Our evidence supports a model in which the dramatic increase in size and phospholipid content of yeast[53], mammalian[54], and plant[55] peroxisomes during high β-oxidation flux allows increased peroxisomal ILV formation to bring fatty acids into the lumen. Indeed, membranes purified from enlarged cotton seedling peroxisomes contain abundant free fatty acids and even some TAG[55]. Moreover, radioactive labeling studies reveal that peroxisomal membrane phospholipids are synthesized from lipid droplet TAG during peroxisome enlargement in cotton seedlings[55].

The use of peroxisomal ILVs to import fatty acids may illuminate the long-standing mystery of why peroxisomes evolved to compartmentalize β-oxidation when mitochondria were already capable of it. Early mitochondria evolved in the aqueous environment of the ocean and might be specialized for importing and catabolizing water-soluble fatty acids. However, insolubility increases with acyl chain length, and the fatty acids stored in eukaryotic TAG are largely water insoluble. The tendency of long-chain fatty acids to partition into lipid bilayers may lessen accessibility to the carnitine shuttle used by mitochondria to import fatty acids across the two lipid bilayers to the catabolic enzymes in the matrix. Moreover, free fatty acids can permeabilize lipid bilayers, particularly if the fatty acid is long enough to interact with the acyl chains of the opposite leaflet[56]. Because oxidative phosphorylation requires high membrane integrity, mitochondria may be particularly sensitive to these membrane-permeabilizing effects. Indeed, mitochondrial dysfunction accompanies peroxisome diseases[46], and peroxisome disruption in hepatocyte models increases mitochondrial β-oxidation[57,58] and impairs inner membrane integrity and membrane potential[59,60].

The ubiquitous peroxisomal membrane complexity revealed here is likely to influence diverse peroxisomal processes, from lumenal and membrane protein targeting to fission and interactions with other organelles, and thus has important implications for both cell biology and the roles of peroxisomes in health and disease. Aberrant peroxisome morphology, including enlarged peroxisomes when β-oxidation enzymes are disrupted[61], is a hallmark of many peroxisomal diseases[62], and peroxisomal subcompartmentalization defects could contribute to the etiology of

human peroxisome biogenesis disorders. Moreover, the ILV scarcity that accompanies impairment of PXA1-mediated fatty acid import may be relevant to defects in X-linked adrenoleukodystrophy, which is caused by a mutated PXA1-related transporter[63]. Peroxisomal ILV defects may also contribute to the etiology of disorders characterized by ESCRT pathway dysfunction, which include a variety of cancers and neurodegenerative disorders[64]. Beyond fatty acids, peroxisomes process other insoluble compounds, including plasmalogens, bile acids, and cholesterol[65]. The use of membrane complexity to handle these molecules may be a general aspect of peroxisome function. By leveraging the advantages of the Arabidopsis system, our findings challenge the long-standing view of peroxisomes as simple single-membrane organelles and may help address fundamental questions in peroxisome function and evolution.

## Methods

**Data reporting**. The investigators were not blinded to allocation during experiments and outcome assessment.

**Plant materials and growth conditions**. Transgenic and mutant lines were in the *Arabidopsis thaliana* Columbia-0 (Col-0) accession. Seeds were surface sterilized in 50% (v/v) consumer bleach supplemented with 0.01% (v/v) Triton X-100, stratified 1 day at 4 °C in the dark, and plated on plant nutrient (PN) medium[66] supplemented with 0.5% (w/v) sucrose (PNS). Unless otherwise noted, seedlings were grown on PNS solidified with 0.6% (w/v) agar on plates wrapped in gas-permeable surgical tape under constant light at 22 °C. Plants used for seed production were transferred to soil after 2–3 weeks and grown under constant light at 22–25 °C.

**DNA cloning methods**. To express mNeonGreen-mPTS$^{PEX26}$ and mRuby3-PTS1 (Supplementary Fig. 1a), we designed a construct encoding mNeonGreen (mNG) fused to the last 90 amino acids of PEX26 (mPTS$^{PEX26}$) driven by the cauliflower mosaic virus (CaMV) *35S* promoter and terminated with the heat shock protein 18.2 (HSP18.2) terminator[67] (flanked by AvrII restriction sites) and mRuby3 fused to the SKL PTS1 sequence driven by the *UBQ10* promoter[68] and terminated by the alcohol dehydrogenase (ADH) terminator[67] (flanked by SpeI restriction sites). The mRuby3 and mNG open reading frames were codon optimized for Arabidopsis using the JCat codon adaptation tool[69], and the insert, flanked by attL sites for use in Gateway cloning (Invitrogen) was synthesized and cloned into the pUC57 vector by GenScript to give pENTRY-mNG-mPTS$^{PEX26}$/mRuby3-PTS1.

To generate constructs encoding a single reporter (Supplementary Fig. 1a), the pENTRY-mNG-mPTS$^{PEX26}$/mRuby3-PTS1 plasmid was digested with SpeI and NheI or AvrII and NheI (which leave compatible 5′ overhangs) and re-ligated to obtain pENTRY-mNG-mPTS$^{PEX26}$ and pENTRY-mRuby3-PTS1, respectively.

To express mPTS$^{PEX22}$-mNG-HA and mRuby3-PTS1 (Supplementary Fig. 1a), we designed a construct encoding the first 94 amino acids of PEX22 (mPTS$^{PEX22}$) fused to the mNG-HA open reading frame and flanked by BstB1 sites (pUC57-mPTS$^{PEX22}$-mNG-HA) for synthesis by GenScript. The BstB1 insert of pUC57-mPTS$^{PEX22}$-mNG-HA was ligated into the BstB1 site of pENTRY-mNG-mPTS$^{PEX26}$/mRuby3-PTS1 in place of mNG-mPTS$^{PEX26}$ to give pENTRY-mPTS$^{PEX22}$-mNG-HA/mRuby3-PTS1.

To generate tri-fluorescent constructs (Supplementary Fig. 1c), we designed an insert flanked by SpeI sites containing the actin 2 (*ACT2*; *At3g18780*) promoter (1000 bp upstream of the initiator codon) separated from the *NOS* terminator by an MfeI site (pUC57-pACT2-MfeI-NOSt). We used Q5 high-fidelity DNA polymerase (NEB) to PCR-amplify (from a construct we designed previously) Arabidopsis codon-optimized (with JCat) mTagBFP2 followed by an MluI site and flanked by MfeI sites. This amplicon was cloned into pENTR/D-TOPO. The mTagBFP2 MfeI insert was ligated into MfeI-cut pUC57-pACT2-MfeI-NOSt vector to give pUC57-pACT2-mTagBFP2-MluI-NOSt. The Act2-mTagBFP2-MluI-NOSt insert was removed with SpeI and ligated into the NheI site of pENTRY-mNG-mPTS$^{PEX26}$/mRuby3-PTS1 or pENTRY-mPTS$^{PEX22}$-mNG/mRuby3-PTS1 to give pENTRY-mNG-mPTS$^{PEX26}$/mTagBFP2-MluI/mRuby3-PTS1 or pENTRY-mPTS$^{PEX22}$-mNG/mTagBFP2-MluI/mRuby3-PTS1. *SCO3* and *UP9* cDNAs were PCR amplified from pENTR223-SCO3 (ABRC clone G50714) and pENTR223-UP9 (ABRC clone G67714) using Q5 high-fidelity DNA polymerase (NEB) and cloned into the MluI site of pENTRY-mNG-mPTS$^{PEX26}$/mTagBFP2-MluI/mRuby3-PTS1 or pENTRY-mPTS$^{PEX22}$-mNG/mTagBFP2-MluI/mRuby3-PTS1 using the Gibson Assembly[70] cloning kit (NEB) to give pENTRY-mNG-mPTS$^{PEX26}$/mTagBFP2-SCO3/mRuby3-PTS1 or pENTRY-mPTS$^{PEX22}$-mNG/mTagBFP2-UP9/mRuby3-PTS1, respectively.

The inserts of various entry vectors (Supplementary Fig. 1a, c) were recombined into the pMCS:GW destination vector[68] using LR Clonase II (Invitrogen) to give the corresponding vectors suitable for plant transformation.

To generate constructs encoding dominant-negative derivatives of SNF7.1 (*At4g29160*) and VPS4/SKD1 (*At2g27600*) (Supplementary Fig. 1b), we used

Gibson assembly with pENTR/SD-TOPO-SNF7 (ABRC clone U15944) and pENTR223-VPS4 (ABRC clone G12244) to introduce the L22W and E232Q mutations and to add C-terminal HA tags, creating pENTR/SD-TOPO-SNF7$^{L22W}$-HA and pENTR223-VPS4$^{E232Q}$-HA. The inserts of these entry vectors were recombined into the β-estradiol induction system plant transformation vector pFZ19 (Addgene plasmid 36184)[71] using LR Clonase II (Invitrogen).

Plasmids were propagated in NEB5α *E. coli* cells (NEB), and inserts were sequenced (LoneStar Labs or GeneWiz, Houston, TX) to ensure the lack of unintended mutations.

**Arabidopsis transformation**. DNA constructs were transformed into *Agrobacterium tumefaciens* GV3101 (pMP90)[72] using electroporation, and the resultant Agrobacterium strains were used to transform wild-type Arabidopsis plants[73]. Transformants expressing fluorescent reporters were selected following growth on PNS plates containing 10 μg/mL Basta. Homozygous, single-insertion lines were identified as lines showing 75% Basta resistance in the second generation (indicating a single-insertion locus) and 100% Basta resistance in the third generation (indicating transgene homozygosity). For fluorescent reporters, lines from at least two independent transformation events were imaged and a representative line was used for all experiments. Transformants for the dominant-negative ESCRT constructs were selected on PNS plates containing 15 μg/mL hygromycin that were incubated in constant light for 1 day followed by 4 days in darkness and 1 day in constant light. Lines exhibiting estradiol-dependent SNF7$^{L22W}$-HA or VPS4$^{E232Q}$-HA accumulation were selected following immunoblotting and were used for estradiol induction experiments.

**Immunoblotting**. Immunoblotting was done as described[74] except that membranes were air dried for 1 h before blocking overnight in 8% (w/v) non-fat dry milk in TBST (Tris-buffered saline with 0.1% [v/v] Tween-20). Primary antibodies used were rabbit anti-PMDH[75] (1:5000), rat anti-HA (1:300, Roche clone 3F10, Sigma 11867423001), and mouse anti-HSC70 (1:50,000, Stressgen SPA-817). Horseradish peroxidase-conjugated secondary antibodies (1:5000 goat anti-rat IgG, Invitrogen A10549; 1:5000 goat anti-rabbit IgG, GenScript A00098; or 1:5000 goat anti-mouse IgG, GenScript A00160) were incubated for 2 h before washing in TBST. Antibodies were imaged using WesternSure Premium Chemiluminescent substrate (LI-COR Biosciences) and a LI-COR Odyssey Fc imaging system. Uncropped blots are included in the Source Data file.

**Confocal microscopy**. For most experiments, fluorescence was imaged using a Carl Zeiss 710 confocal microscope equipped with Zen 2010 software (version 6.0.0.485), a Meta detector and plan-apochromat ×20/0.8 air/dry, ×63/1.4 oil-immersion, and ×100/1.4 oil-immersion objectives. mNeonGreen, mRuby3, and mTagBFP2 were excited with 488, 543, and 405 nm lasers, respectively. Alternatively, fluorescence was imaged using a Carl Zeiss 800 confocal microscope equipped with Zen 2.6 software (version 2.6.76.00), a Meta detector and plan-apochromat ×20/0.8 air/dry, ×63/1.4 water-immersion, and ×100/1.4 oil-immersion objectives or an Olympus/Andor spinning disk confocal microscope equipped with Andor iQ software (version 3.4) with a ×100 oil-immersion objective. Images were processed in ImageJ/Fiji (version 2.0.0)[76]. 3D projections were acquired by imaging from ~10 to 20 μm below the top surface of the epidermal cell layer to the top surface of the mesophyll cells and creating a z-projection in ImageJ from part or all of the resulting z-stack.

To visualize peroxisomes with mPTS$^{PEX26}$ in 8-day-old seedlings (Fig. 1a, c, d), cotyledons of seedlings expressing mNeonGreen-mPTS$^{PEX26}$ and mRuby3-PTS1 (Supplementary Fig. 1a) were imaged using a ×63 objective with 12-bit depth and a two-track imaging setup, with imaging tracks switching every line. mNeonGreen fluorescence was captured at 493–579 nm with an airy unit of 1.58, corresponding to a 1.3 μm optical slice. mRuby3 fluorescence was captured at 582–651 nm with an airy unit of 1.38, corresponding to a 1.2 μm optical slice.

To visualize peroxisomes with mPTS$^{PEX22}$ in 8-day-old seedlings (Fig. 1b, e, f), cotyledons of seedlings expressing mNeonGreen-mPTS$^{PEX22}$ and mRuby3-PTS1 (Supplementary Fig. 1a) were imaged. mNeonGreen and mRuby3 were imaged using a ×100 objective on the same track with an airy unit of 1, corresponding to a 0.8 μm optical slice. mNeonGreen fluorescence was captured at 494–533 nm, and mRuby3 fluorescence was captured at 620–657 nm. Each image is an average of two images with 12-bit depth.

To visualize peroxisomal ILVs with mPTS$^{PEX26}$ in 4-day-old seedlings (Fig. 1g, h, m), whole seedlings expressing mNeonGreen-mPTS$^{PEX26}$ and mRuby3-PTS1 (Supplementary Fig. 1a) were imaged. Seedlings were imaged using a ×20 or ×63 objective with 12-bit depth and a two-track imaging setup, with imaging tracks switching every line. mNeonGreen fluorescence was captured at 490–558 nm with an airy unit of 1.16, corresponding to a 0.8 μm optical slice. mRuby3 fluorescence was captured at 580–650 nm with an airy unit of 0.99, corresponding to a 0.9 μm optical slice.

To visualize peroxisomal ILVs with mPTS$^{PEX22}$ in 3-day-old seedlings (Fig. 1i, j), whole seedlings expressing mPTS$^{PEX22}$-mNeonGreen-HA and mRuby3-PTS1 (Supplementary Fig. 1a) were imaged. mNeonGreen and mRuby3 were imaged using a ×100 objective on the same track with an airy unit of 1, corresponding to a 0.8 μm optical slice. mNeonGreen fluorescence was captured at 494–533 nm, and

mRuby3 fluorescence was captured at 618–657 nm. Each image is an average of two images with 12-bit depth.

For FM4-64 experiments (Fig. 1l), 4-day-old seedlings expressing mNeonGreen-mPTS$^{PEX26}$ (Supplementary Fig. 1a) were stained with 3 mL of 5 μM FM4-64 in water for 2 h in a six-well plate, washed twice with 5 mL of H$_2$O, and imaged using a ×63 objective. Seedlings were imaged using a two-track imaging setup, with imaging tracks switching every line. mNeonGreen and FM4-64 were excited with a 488 nm laser. mNeonGreen fluorescence was captured at 493–567 nm with an airy unit of 1.01, corresponding to a 0.8 μm optical slice, and FM4-64 fluorescence was captured at 679–759 nm with an airy unit of 1.01, corresponding to an 0.8 μm optical slice. Each image is an average of two images with 12-bit depth.

To visualize peroxisomal ILVs with mRuby3-PTS1 (Fig. 1n), seedlings expressing mRuby3-PTS1 (Supplementary Fig. 1a) were imaged using a ×100 objective and 12-bit depth. mRuby3 fluorescence was captured at 550–675 nm, with an airy unit of 1, corresponding to a 0.9 μm optical slice.

For time-lapse imaging (Fig. 2), we modified a previously described long-term imaging protocol[13]. Cotyledons of 4-day-old light-grown seedlings expressing mNeonGreen-mPTS$^{PEX26}$ and mRuby3-PTS1 were removed and placed on a μ-Dish 35-mm high well (Ibidi 81151) (Fig. 2a) or a 1-well Lab-Tek II chambered coverglass (ThermoFisher 155360) (Fig. 2b) under a 2 mL slab of 1% (w/v) agar in water to maintain moisture and allow gas exchange. For Fig. 2a, cotyledons were imaged using a ×100 objective on an Olympus/Andor spinning disk confocal microscope, exciting green (488 nm) and red (561 nm) fluorescence on different tracks switching every frame with an airy unit of 1. Z-stacks were acquired using filter cubes for green (TR-F525-030 Single Band Fluorescence Filter, 525 Center) and red (TR-F607-036 Single Band Fluorescence Filter, 607 Center) emission every 15 min over 10 h, and each image is an average of four images with 16-bit depth. For Fig. 2b, cotyledons were imaged using a Carl Zeiss 710 confocal microscope with a ×20 objective and a two-track imaging setup, with imaging tracks switching every line. mNeonGreen fluorescence was captured at 490–558 nm with an airy unit of 1.17, corresponding to a 2.1 μm optical slice. mRuby3 fluorescence was captured on a separate imaging track at 580–650 nm, with an airy unit of 1, corresponding to a 2.1 μm optical slice. Z-stacks were acquired every 15 min over 10 h, and each image is an average of two images with 12-bit depth.

For imaging peroxisomes after dominant-negative ESCRT induction (Figs. 3, 5 and Supplementary Fig. 2), we crossed homozygous mNeonGreen-mPTS$^{PEX26}$ mRuby3-PTS1 plants to pXVE-SNF7$^{L22W}$-HA or pXVE-VPS4$^{E232Q}$-HA plants. The subsequent homozygous lines were sown directly into liquid PNS media with 25 μM β-estradiol (from a 50 mM stock in DMSO) or the equivalent amount of DMSO (mock) and incubated in six-well plates under continuous light for 5 days with gentle shaking. mNeonGreen and mRuby3 were imaged in cotyledons using a ×20 objective with 8-bit depth on different tracks with the tracks switching every line. mNeonGreen fluorescence was captured at 490–558 nm with an airy unit of 1.17, corresponding to a 2.4 μm optical slice. mRuby3 fluorescence was captured at 580–650 nm with an airy unit of 1, corresponding to a 2.1 μm optical slice.

For imaging peroxisomes in β-oxidation mutants (Fig. 4), we introduced mNeonGreen-mPTS$^{PEX26}$ mRuby3-PTS1 into previously described mutants (pxn-4[9], acx2-2[9], mfp2-8[9], pxa1-1[22], and pxn-4 pxa1-1[9]) by crossing homozygous mNeonGreen-mPTS$^{PEX26}$ mRuby3-PTS1 plants to homozygous mutants. mNeonGreen and mRuby3 were imaged in cotyledons from age-matched seedlings using a Zeiss 800 confocal microscope and a ×63 water objective with 16-bit depth on different tracks with the tracks switching every line. mNeonGreen fluorescence was captured at 410–570 nm with an airy unit of 1.24, corresponding to a 0.6 μm optical slice. mRuby3 fluorescence was captured at 570–695 nm with an airy unit of 1, corresponding to a 0.7 μm optical slice.

For visualizing lipid droplets (Fig. 5), seedlings were stained with 1 mL of 100 μM monodansylpentane (MDH) in 50 mM Tris-HCl (pH 7.5) for at least 30 min in 1.5 mL tube prior to imaging using a Zeiss 800 confocal microscope with a ×63 water objective. MDH was excited with a 405 nm laser, mNeonGreen was excited with a 488 nm laser, and mRuby3 was excited with a 561 nm laser. MDH, mNeonGreen, and mRuby3 were imaged on different tracks—MDH and mRuby3 on one track with an airy unit of 1, corresponding to a 0.5 μm optical slice, and mNeonGreen was imaged on the other track with an airy unit of 0.92, corresponding to a 0.5 μm optical slice and the tracks switched every line. MDH and mNeonGreen fluorescence was captured at 410–561 nm and mRuby3 fluorescence was captured at 561–605 nm. Each image is an average of one or two images with 16-bit depth.

For imaging SCO3 and UP9 with peroxisome membrane and lumen reporters (Fig. 6b–e), seedlings expressing mRuby3-PTS1, mNeonGreen-mPTS$^{PEX26}$ or mPTS$^{PEX22}$-mNeonGreen, and mTagBFP2-SCO3 or mTagBFP2-UP9 (Supplementary Fig. 1c) were imaged using a Zeiss 800 confocal microscope with a ×63 water objective. mTagBFP2 was excited with a 405 nm laser, mNeonGreen was excited with a 488 nm laser, and mRuby3 was excited with a 561 nm laser. mNeonGreen, mTagBFP2, and mRuby3 were imaged on different tracks—mTagBFP2 and mRuby3 on one track with an airy unit of 1, corresponding to a 0.5 μm optical slice, and mNeonGreen was imaged on the other track with an airy unit of 0.92, corresponding to a 0.5 μm optical slice and the tracks switched every line. mTagBFP2 and mNeonGreen fluorescence was captured at 410–561 nm and mRuby3 fluorescence was captured at 605–700 nm. Each image is an average of two images with 16-bit depth.

**Image quantification, statistical analysis, and reproducibility**. To quantify peroxisome size, z-projections of the mNeonGreen and mRuby3 channel merge of z-stack images from three (or two, as noted by * in Fig. 4b) cotyledons (from different plants) for each condition were made in ImageJ (version 2.0.0) and imported into Ilastik (version 1.3.2)[77] to create binary masks (Supplementary Fig. 2). Binary masks were made by training the pixel classification module to both the uninduced and induced images from each line (Fig. 3) or to wild-type images (Fig. 4). The masks were imported into ImageJ for final processing: the fill holes function was used to fill any holes in the peroxisome masks, the close function was used for background subtraction, and the watershed function was used to separate touching objects. Peroxisome cross-sectional area was then measured in ImageJ for each mask, excluding objects <1 μm$^2$ or touching an edge of the image; area measurements were then converted to diameter in Microsoft Excel (version 2010). Quantifying lipid droplet size was done similarly using only the lipid droplet channel for training and two cotyledons (from different plants) for each condition.

Organelle measurements (provided in the Source Data file) were analyzed using R (version 3.5.3). Student's two-tailed unpaired t tests were used to compare organelle size with and without β-estradiol. Boxplots were generated using the ggplot2 data visualization package (version 3.3.2)[78].

Experiments were repeated at least twice with similar results, and micrographs from representative experiments are shown. Figures were assembled using Adobe Illustrator (version 24.2.3).

**Reporting summary**. Further information on research design is available in the Nature Research Reporting Summary linked to this article.

## Data availability

The data that support the findings of this study are available in the main text or supplementary information. All unique biological materials (e.g., plant lines, DNA constructs) are available from the corresponding author upon request. Source data are provided with this paper.

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

## Acknowledgements

We thank Lucia Strader for the pMCS:GW vector, the ABRC for cDNA clones, Aryeh Warmflash for spinning disk microscope use, and Gabrielle Buck, Roxanna Llinas, DurreShahwar Muhammad, Dereth Phillips, Kathryn Smith, Ana Swearingen, Melissa Traver, and Andrew Woodward for feedback on the manuscript. This work was supported by the National Institutes of Health (NIH R01GM079177 and R35GM130338) and the Robert A. Welch Foundation (C-1309). Confocal equipment was obtained through a Shared Instrumentation grant (NIH S10RR026399) and supported by the Shared Equipment Authority at Rice University.

## Author contributions

Z.J.W. and B.B. designed the experiments, Z.J.W. performed the experiments, and Z.J.W. and B.B. wrote the manuscript.

## Competing interests

The authors declare no competing interests.
