## [Peer Review File · Nature Communications]

Reviewers' comments:

Reviewer #1 (Remarks to the Author):

Wright and Bartel report internal membranes in Arabidopsis seedling peroxisomes. Peroxisomal intraluminal vesicles (ILVs) were most apparent in early 4-day-old seedlings. They were visualized in organelles tagged with fluorescent protein-conjugated targeting domains from the membrane proteins PEX26 and PEX22. As the seeds mature, peroxisomes became smaller and more densely filled with ILVs. Mechanistically, the authors showed that disruption of the ESCRT pathway resulted in fewer ILVs and larger organelles, while disruption of autophagy had no effect. No engulfment of lipid droplets were observed. Interestingly, the internal compartments of peroxisomes were not homogeneous, with some matrix proteins partitioning in some of the ILVs.

While there have been preliminary reports of intraperoxisomal membranes in the past (cited in the Discussion), this report offers the most convincing example of peroxisomal ILVs. The fluorescent micrographs are beautiful. Controls are performed to rule out artifacts caused by expression of the fluorescent membrane probes.

While this manuscript presents compelling evidence regarding the presence of peroxisomal ILV, there is no information regarding function of these internal compartments and only limited information regarding composition of the ILVs, biogenesis of the subcompartments, or morphogenesis into simpler structures during development. Why are ILVs there? This may be too hard a question to answer with current technology. Is so, more information should be provided into mechanism of ILV formation or their conversion to more typical peroxisomes later in life.

Other comments:

(1) Electron micrographs and immuno-gold (with anti-SKL antibody, for example) would complement well the light microscopic images, especially at 8 days after germination when internal membranes are difficult to see (Fig. 1).

(2) Quantification should be provided to back up the statement that peroxisomes get smaller over time with more ILV membranes. While organellar size data are provided, it is not convincing that the smaller size is accompanied by more intraperoxisomal membrane (rather than the same amount of membrane compacted into a smaller space). Electron microscopy may be useful here.

(3) Lines 77-78. Fig. 1b hardly shows internal membranes, as stated. Did the authors mean Fig. 1e?

Reviewer #2 (Remarks to the Author):

This manuscript shows that in young seedlings expressing fluorescently tagged peroxisome markers, hypocotyl cells contain very large peroxisomes that develop internal vesicles. The authors used the targeting peptides of two peroxins (PEX26 and PEX22) to insert mNeonGreen to the peroxisomal membrane, resulting in two peroxisome membrane reporters. They co-expressed these reporters with mRuby3-PST1, which is delivered to the peroxisome lumen. Confocal imaging of these reporters showed that peroxisomes in young seedlings are exceptionally large and contain large vesicles in their lumen. For example, the peroxisome depicted in Fig 1h is over 15 microns in diameter, this is approximately two times larger than a fully developed chloroplast and 1.5 times larger than a nucleus. The intraluminal vesicles are several microns in diameter. By expressing the same markers in an autophagy mutant, the authors ruled out that these unusual organelles are autophagosomes; based on their inability to take the endocytic tracker FM4-64, they conclude that they are not endosomes, another organelle that develops intraluminal vesicles. Since

endosomes require ESCRT proteins to form their vesicles, the authors expressed two dominant negative forms of ESCRT proteins under an inducible promoter to test whether ESCRTs are also needed for the formation peroxisome inner membranes. They conclude that, based on the observed enlargement of peroxisome size and reduction of intraperoxisomal membranes, ESCRTs are needed for the formation of inner vesicles in plant peroxisomes.

This manuscript reports a potentially novel aspect of peroxisome morphology and function in plants and as such, I find it extremely interesting. However, in its current form, it doesn't provide enough experimental evidence to support some of the authors' claims. Plant peroxisomes are usually 1-2 microns in diameter. The authors explain that old electron microscopy images show evidence that some plant endosomes may contain internal membrane. Although this might be case, this manuscript is exclusively based on gigantic peroxisomes (10 times larger than a regular peroxisome) that have never been reported before. I appreciate that they are using different fluorescent markers and obtaining the same results but do these unusual peroxisomes exist in the absence of any overexpressed peroxisome marker?

I also find the claim that ESCRT proteins are needed to form peroxisome inner membranes weak. The authors see and quantify a change in the area of peroxisomes (they become larger) when they express the dominant-negative ESCRTs. However, this is rather indirect and could be due to other functions that ESCRT are known to regulate like autophagy. The authors also mention that they see less inner peroxisome membrane under these conditions but they do not quantify this phenomenon. In addition, I would argue that it is unreliable to precise the amount of intraluminal membranes by confocal microscopy as they indicate that not all peroxisome inner compartments contain the same markers (Fig 5) and their peroxisome lumen marker would fail to reliably report concentric inner membranes like those observed in Fig 4a. There are several viable ESCRT mutants characterized in plants, do they have abnormal peroxisomes with no inner membranes?

If the authors could resolve these two main issues, this could be a high impact publication in the cell biology field.

Reviewer #1

Wright and Bartel report internal membranes in Arabidopsis seedling peroxisomes. Peroxisomal intraluminal vesicles (ILVs) were most apparent in early 4-day-old seedlings. They were visualized in organelles tagged with fluorescent protein-conjugated targeting domains from the membrane proteins PEX26 and PEX22. As the seeds mature, peroxisomes became smaller and more densely filled with ILVs. Mechanistically, the authors showed that disruption of the ESCRT pathway resulted in fewer ILVs and larger organelles, while disruption of autophagy had no effect. No engulfment of lipid droplets were observed. Interestingly, the internal compartments of peroxisomes were not homogeneous, with some matrix proteins partitioning in some of the ILVs.

While there have been preliminary reports of intraperoxisomal membranes in the past (cited in the Discussion), this report offers the most convincing example of peroxisomal ILVs. The fluorescent micrographs are beautiful. Controls are performed to rule out artifacts caused by expression of the fluorescent membrane probes.

While this manuscript presents compelling evidence regarding the presence of peroxisomal ILV, there is no information regarding function of these internal compartments and only limited information regarding composition of the ILVs, biogenesis of the subcompartments, or morphogenesis into simpler structures during development. Why are ILVs there? This may be too hard a question to answer with current technology. Is so, more information should be provided into mechanism of ILV formation or their conversion to more typical peroxisomes later in life.

Response: We agree that the functional question is fascinating to contemplate and difficult to address. A prerequisite for understanding ILV function(s) will be to obtain mutants that fail to efficiently form these vesicles. To begin this process, we began surveying peroxisome-defective mutants for their ability to form ILVs. Intriguingly, we found that the large peroxisomes of *pxn*, *acx2*, and *mfp2* mutant have few ILVs, implicating these proteins in efficient ILV formation. Moreover, rather than being freely floating the lumen in wild-type peroxisomes, the ILVs in these mutants appeared to be tethered to the outer membrane, as if ILV formation, when it did occur, was not complete. We further found that the *pxa1* mutant, which is defective in the transporter that brings fatty acids into the peroxisome, has dramatic defects in ILV formation that are epistatic to *pxn* defects. The roles identified for PXN, ACX2, and MFP2 could be direct (e.g., as a target on the peroxisome membrane for ESCRT activity or as an ILV stabilizing protein in the lumen) or indirect (e.g., due to a metabolic role). We include these new data in a new Figure 4.

We extended these functional studies by expanding our examination of lipid droplet-peroxisome ILV relationships. We found that ESCRT inhibition not only impaired peroxisomal ILV formation (Figure 3), but it also impaired lipid droplet mobilization (new Figure 5a, b). Moreover, we found that the retained lipid droplets in *acx2*, for example, appear directly apposed to the aberrant sites of ILV formation (new Figure 5c). These findings bolster the possibility that ILVs play a functional role in lipid metabolism.

We expand on the implications of these new findings for understanding functional roles of ILVs and the evolution of peroxisomes in the revised discussion. We hope that our report of these

peroxisomal ILVs will enable us and other peroxisome researchers to consider further functional questions as additional technologies are developed.

Other comments:

(1) Electron micrographs and immuno-gold (with anti-SKL antibody, for example) would complement well the light microscopic images, especially at 8 days after germination when internal membranes are difficult to see (Fig. 1).

Response: We have tried a variety of TEM experiments and have obtained some intriguing images that may include peroxisomes. Unfortunately, this technique suffers from low throughput and it is difficult to consistently visualize membranes, particularly when protein levels are high (as in the peroxisome lumen). The low throughput presents a particular challenge, because although the eye is drawn to the large peroxisomes with clear ILVs in our confocal images, our quantification of these images (e.g., Figure 3d) reveals that these large organelles are the minority and most peroxisomes are small and densely packed. These limitations in combination with the heterogeneity of peroxisome morphologies that are apparent in confocal images preclude us from providing TEM images that we are confident are representative examples.

(2) Quantification should be provided to back up the statement that peroxisomes get smaller over time with more ILV membranes. While organellar size data are provided, it is not convincing that the smaller size is accompanied by more intraperoxisomal membrane (rather than the same amount of membrane compacted into a smaller space). Electron microscopy may be useful here.

Response: In response to this suggestion, we have supplemented the time-course studies of Figure 2 by repeating the experiment using a spinning disc confocal microscope that allowed higher magnification. We provide a new Figure 2a (and accompanying movie) that allows a more clear view of individual peroxisomes as they shrink. These new images clearly show additional vesicles accumulating within several individual peroxisomes over time. We currently do not have a method to reliably quantify ILV numbers given their differing protein compositions, sizes, and densities. In the absence of a reliable method for quantifying inner membrane density, we have tempered our language to ensure that we are clearly stating that our data are consistent with a model in which ILVs derive from the outer membrane.

(3) Lines 77-78. Fig. 1b hardly shows internal membranes, as stated. Did the authors mean Fig. 1e?

Response: We thank the reviewer for pointing out this typo; we intended to refer to Fig. 1c here.

Reviewer #2

This manuscript shows that in young seedlings expressing fluorescently tagged peroxisome markers, hypocotyl cells contain very large peroxisomes that develop internal vesicles. The authors used the targeting peptides of two peroxins (PEX26 and PEX22) to insert mNeonGreen to the peroxisomal membrane, resulting in two peroxisome membrane reporters. They co-expressed these reporters with mRuby3-PST1, which is delivered to the peroxisome lumen. Confocal imaging of these reporters showed that peroxisomes in young seedlings are exceptionally large and contain large vesicles in their lumen. For example, the peroxisome depicted in Fig 1h is over 15 microns in diameter, this is approximately two times larger than a fully developed chloroplast and 1.5 times larger than a nucleus. The intraluminal vesicles are several microns in diameter. By expressing the same markers in an autophagy mutant, the authors ruled out that these unusual organelles are autophagosomes; based on their inability to take the endocytic tracker FM4-64, they conclude that they are not endosomes, another organelle that develops intraluminal vesicles. Since endosomes require ESCRT proteins to form their vesicles, the authors expressed two dominant negative forms of ESCRT proteins under an inducible promoter to test whether ESCRTs are also needed for the formation of peroxisome inner membranes. They conclude that, based on the observed enlargement of peroxisome size and reduction of intraperoxisomal membranes, ESCRTs are needed for the formation of inner vesicles in plant peroxisomes.

(1) This manuscript reports a potentially novel aspect of peroxisome morphology and function in plants and as such, I find it extremely interesting. However, in its current form, it doesn't provide enough experimental evidence to support some of the authors' claims. Plant peroxisomes are usually 1-2 microns in diameter. The authors explain that old electron microscopy images show evidence that some plant endosomes may contain internal membrane. Although this might be case, this manuscript is exclusively based on gigantic peroxisomes (10 times larger than a regular peroxisome) that have never been reported before. I appreciate that they are using different fluorescent markers and obtaining the same results but do these unusual peroxisomes exist in the absence of any overexpressed peroxisome marker?

Response: We agree that the subpopulation of unusually large peroxisomes transiently present in young seedlings have facilitated our initial observation of and subsequent characterization of peroxisomal internal vesicles. The dynamic size of Arabidopsis seedling peroxisomes was previously reported by Rinaldi et al. (*Genetics* 2016, 204:1089-1115), and because this size dynamism is not well known in the community of researchers, we have quantified these size changes in our various genotypes in a new Figure 4. Furthermore, peroxisome enlargement during high beta-oxidation conditions is observed in yeast and mammalian cells and we are mostly imaging Arabidopsis during times of peak beta-oxidation. Our imaging in older seedlings after beta-oxidation of seed stores is complete (e.g., Fig. 1a, 4a) reveals peroxisome sizes similar to what is commonly observed in plants. Additionally, our imaging in root cells (Fig. 4d) shows small peroxisomes with or without ILVs in wild-type and *pxa1-1*, respectively, further supporting the conclusion that inner membranes are not just a feature of the largest peroxisomes.

Indeed, our quantitative data in Fig. 4b reinforce the finding that even though the eye is drawn to the larger peroxisomes in Fig. 4a, most peroxisomes are indeed 1-2 μm in diameter (as

expected by the reviewer) even when larger peroxisomes are relatively abundant. This low frequency increases the difficulty of detecting peroxisome size changes using low throughput methods that lack reliable identifying markers and are more difficult to quantify without bias (e.g., via TEM).

We also note that most plant researchers use luminal markers (e.g., GFP-PTS1) to observe peroxisomes in plants, and that most of the luminal markers in the literature are early generation fluorescent protein (e.g., GFP). Although the transient larger peroxisomes of seedlings can be detected with these markers (e.g., Rinaldi et al., 2016), the limited brightness of these markers (compared to the brighter reporters that we employ here), combined with the possibly more dilute lumen in larger peroxisomes, may explain why more researchers have not remarked upon the transient large peroxisomes that we detect in young seedlings. Our new reporters reduce but do not completely eliminate this sensitivity problem, both by employing brighter fluorescent proteins and also by labeling the membrane (which may be less subject to dilution in large peroxisomes). For example, the large peroxisome in the upper left quadrant of Figure 1a and c is less apparent than the smaller, more densely packed peroxisomes in the same image.

(2) I also find the claim that ESCRT proteins are needed to form peroxisome inner membranes weak. The authors see and quantify a change in the area of peroxisomes (they become larger) when they express the dominant-negative ESCRTs. However, this is rather indirect and could be due to other functions that ESCRT are known to regulate like autophagy. The authors also mention that they see less inner peroxisome membrane under these conditions but they do not quantify this phenomenon. In addition, I would argue that it is unreliable to precisely measure the amount of intraluminal membranes by confocal microscopy as they indicate that not all peroxisome inner compartments contain the same markers (Fig 5) and their peroxisome lumen marker would fail to reliably report concentric inner membranes like those observed in Fig 4a. There are several viable ESCRT mutants characterized in plants, do they have abnormal peroxisomes with no inner membranes?

Response: As suggested, we have used several approaches to strengthen our conclusion that the ESCRT proteins contribute to peroxisomal ILV formation. We repeated the ESCRT dominant negative induction experiments using lines homozygous for the transgenes (rather than the hemizygous F1 lines used in the original submission). Having made a connection to lipid mobilization, we also measured peroxisome size in younger seedlings, closer to the time when lipid stores are depleted in wild type, to maximize any differences. Indeed, we find significantly increased peroxisome diameter when ESCRT is impeded and have included these new data in a new Fig. 3c and d.

Although it is formally possible that the role for ESCRT in peroxisomal ILV formation is indirect, we have ruled out the possibility that such an indirect role is mediated through a role for ESCRT in autophagy. We find that even a mutant completely blocked in autophagy, *atg7-4*, makes peroxisomal ILVs that are indistinguishable from autophagy-competent lines (compare Fig. 1g and 1m).

We agree with the reviewer that it could be interesting to examine our new peroxisomal markers in viable ESCRT mutants. In fact, when we first observed peroxisomal ILVs and wondered about ESCRT involvement, we searched for viable mutants with which to test this hypothesis. It is difficult to predict, however, which allele(s) of which gene(s) might be fruitful for this analysis. All of the core ESCRT-III components (VPS2, VPS20, VPS24, and SNF7) are encoded by at least two genes in Arabidopsis (Gao et al., 2017, Trends Plant Sci 22:986-998). Based on our understanding of the literature, the null allele of VPS2-1 is embryo lethal, and the functions of the remaining genes have only been explored via overexpression of dominant negative versions (as we employ for SNF7.1 in our manuscript). Although *VPS4* is a single-copy gene in Arabidopsis, our understanding is that the *vps4* (*skd1*) dominant-negative constructs (which we employ) were developed for plants because no viable *vps4* alleles are available. Like the core ESCRT-III components, the ESCRT-III accessory protein CHMP1 is encoded by two genes in Arabidopsis; the single mutants lack notable phenotypes and the double mutant is embryo or seedling lethal. As a result, we chose to employ the dominant-negative strategy (with two genes acting at different steps in ILV formation) that has been informative in most studies of ESCRT-III function in Arabidopsis. In summary, we are not enthusiastic about crossing our reporter lines into various viable ESCRT mutants because the viable single mutants of which we are aware lack notable phenotypes (presumably due to redundancy). Moreover, even if we found abnormal peroxisomes in these mutants, such a finding would not clarify whether the role for ESCRT in peroxisomal ILV formation was direct or indirect, and thus would not provide a distinct line of evidence for ESCRT involvement beyond the evidence that we have obtained using the dominant negative SNF7 and VPS4 derivatives.

Finally, we acknowledge that we have not provided experimental evidence that would distinguish whether the role for ESCRT we have identified is direct or indirect, and we have edited the manuscript to indicate that further experiments (e.g., the identification of ESCRT-interacting partners on the peroxisome membrane) will be needed to conclusively determine if the role that ESCRT plays in ILV formation is direct.

If the authors could resolve these two main issues, this could be a high impact publication in the cell biology field.

Response: We appreciate the insightful and constructive comments of both reviewers. We believe that with the new data presented in Fig. 2a, Fig. 4, and Fig. 5, the improved images in Fig. 3c and Fig. 6b-e, and numerous clarifications to the text, the manuscript is considerably strengthened. We are happy to discuss any of these issues further.

REVIEWERS' COMMENTS

Reviewer #1 (Remarks to the Author):

In this revision, Wright and Bartel flesh out their observation that the ESCRT pathway is responsible for peroxisomal intraluminal vesicles (ILVs). In new data they show that the absence of beta-oxidation proteins prevents these vesicles and by showing that neutral lipids in lipid droplets accumulate in the ESCRT mutants. There is heterogeneity in peroxisomal protein targeting to the structures. They also have expanded the Discussion to include hypotheses about the function of ILVs, targeting the import of long-chain fatty acids by ILV membranes.

I found this revision improves the manuscript. The large LDs in the ESCRT mutants are consistent with a role of ILVs in fatty acid oxidation, and I like the hypothesis that the ILVs are there to import insoluble long-chain fatty acids. It is all very intriguing and exciting. What is still missing, in my opinion, is a direct demonstration that the ILVs are important for fatty acid import or oxidation. My suggestions (the authors may have alternatives) to solve this issue while taking minimal additional time for experimentation is performing one of the following in their ESCRT mutant:

- (1) Inhibition of TAG biosynthesis (cerulenin) showing that the large LDs are due to lower lipolysis rather than increased synthesis of neutral lipid.
- (2) Preliminary lipidomics showing a lack of fatty acid oxidation intermediates. Poisoning mitochondrial respiration may be necessary to see an effect.
- (3) Decrease in mitochondrial CO₂ production, which could be attributed to lower peroxisomal acetyl-CoA production from fatty acids.

Besides that criticism, I am very happy with the revision.

Reviewer #2 (Remarks to the Author):

In the revision, the authors have addressed my previous comments. I don't have further suggestions to make.

Reviewer #1:

In this revision, Wright and Bartel flesh out their observation that the ESCRT pathway is responsible for peroxisomal intraluminal vesicles (ILVs). In new data they show that the absence of beta-oxidation proteins prevents these vesicles and by showing that neutral lipids in lipid droplets accumulate in the ESCRT mutants. There is heterogeneity in peroxisomal protein targeting to the structures. They also have expanded the Discussion to include hypotheses about the function of ILVs, targeting the import of long-chain fatty acids by ILV membranes.

I found this revision improves the manuscript. The large LDs in the ESCRT mutants are consistent with a role of ILVs in fatty acid oxidation, and I like the hypothesis that the ILVs are there to import insoluble long-chain fatty acids. It is all very intriguing and exciting. What is still missing, in my opinion, is a direct demonstration that the ILVs are important for fatty acid import or oxidation. My suggestions (the authors may have alternatives) to solve this issue while taking minimal additional time for experimentation is performing one of the following in their ESCRT mutant:

- (1) Inhibition of TAG biosynthesis (cerulenin) showing that the large LDs are due to lower lipolysis rather than increased synthesis of neutral lipid.
- (2) Preliminary lipidomics showing a lack of fatty acid oxidation intermediates. Poisoning mitochondrial respiration may be necessary to see an effect.
- (3) Decrease in mitochondrial CO₂ production, which could be attributed to lower peroxisomal acetyl-CoA production from fatty acids.

Besides that criticism, I am very happy with the revision.

Response:

Thank you for your time. We agree that determining whether the peroxisomal ILVs are acting directly in fatty acid import or oxidation is a critical next step. We agree that the experiments suggested by the reviewer could, if successful, provide additional supportive evidence for this hypothesis. However, given the diverse roles of the ESCRT machinery in the cell, we believe that even the suggested experiments would not demonstrate a direct role for ILVs in fatty acid import or oxidation. We hope that given the tools and observations presented in the current manuscript, we or others will be able to validate or refute such a direct role in the future, perhaps by developing new visualization tools or an in vitro reconstitution system.

Reviewer #2:

In the revision, the authors have addressed my previous comments. I don't have further suggestions to make.

Response:

Thank you for your time.